# No Time to Die: How Kidney Cancer Evades Cell Death

**DOI:** 10.3390/ijms23116198

**Published:** 2022-05-31

**Authors:** Carlo Ganini, Manuela Montanaro, Manuel Scimeca, Giampiero Palmieri, Lucia Anemona, Livia Concetti, Gerry Melino, Pierluigi Bove, Ivano Amelio, Eleonora Candi, Alessandro Mauriello

**Affiliations:** 1Department of Experimental Medicine, TOR, University of Rome Tor Vergata, 00133 Rome, Italy; carlo.ganini@uniroma2.it (C.G.); manuela.montanaro@uniroma2.it (M.M.); manuel.scimeca@uniroma2.it (M.S.); giampiero.palmieri@uniroma2.it (G.P.); anemona@uniroma2.it (L.A.); livia.concetti@students.uniroma2.eu (L.C.); melino@uniroma2.it (G.M.); pierluigi.bove@uniroma2.it (P.B.); ivano.amelio@uniroma2.it (I.A.); candi@uniroma2.it (E.C.); 2Biochemistry Laboratory, Istituto Dermopatico Immacolata (IDI-IRCCS), 00100 Rome, Italy

**Keywords:** renal cell carcinoma, cell death, apoptosis, ferroptosis, pyroptosis, necroptosis

## Abstract

The understanding of the pathogenesis of renal cell carcinoma led to the development of targeted therapies, which dramatically changed the overall survival rate. Nonetheless, despite innovative lines of therapy accessible to patients, the prognosis remains severe in most cases. Kidney cancer rarely shows mutations in the genes coding for proteins involved in programmed cell death, including p53. In this paper, we show that the molecular machinery responsible for different forms of cell death, such as apoptosis, ferroptosis, pyroptosis, and necroptosis, which are somehow impaired in kidney cancer to allow cancer cell growth and development, was reactivated by targeted pharmacological intervention. The aim of the present review was to summarize the modality of programmed cell death in the pathogenesis of renal cell carcinoma, showing in vitro and in vivo evidence of their potential role in controlling kidney cancer growth, and highlighting their possible therapeutic value.

## 1. Introduction

In recent years, an increase in the incidence of cancers was observed worldwide [1], due to an increase in the aging population. In particular, renal cell carcinoma (RCC) showed an increment of around 20% in its incidence over the last 10 years, especially in industrialized countries [1]. Despite the introduction of new and reliable diagnostic analysis, as well as innovative therapies, the mortality rate of RCC remains too high. This is probably due to the presence of multiple genetic and epigenetic alterations underlying the molecular mechanisms that makes RCCs resistant to several therapies. Frequently, these alterations influence the complex signaling network involved in both initiating and carrying out programmed cell death. In fact, an important role is played by the aberrant overexpression of anticancer cell death proteins (e.g., survivin/BIRC5). In this context, the use of nivolumab, an antibody against programmed cell death 1 (PD-1), seems to open a new way to RCC treatment [2].

However, not all kidney tumors show the same prognosis. Over the years, new subtypes were added to the classification of RCCs [3]. The last WHO classification of tumors of the urinary system includes different subtypes of RCCs, according to predominant histological features and to pathognomonic molecular alterations, i.e., the Mit family translocation RCC, and the succinate dehydrogenase-deficient RCC. However, the incidence of the last two subtypes is very low [3]. The great majority of renal cells carcinoma show non-specific molecular signatures.

Clear cell RCC is the major histological subtype, accounts for 65–70% of cases, and originates from the proximal tubule epithelial cells [1]. The prognosis depends on pathological stage, histological grading, presence of necrosis, and sarcomatoid or rhabdoid differentiation [3]. The second most frequent subtype of RCC, papillary renal cell carcinoma, also originates from the cells of the proximal convoluted tubule, accounting for about 18–20% of all cases. Papillary renal cell carcinomas are divided in type 1 and type 2, according to the morphological aspects of the papillae, even if specific genetic profiles can be distinguished among the variants [3].The other most frequent subtypes of renal cell carcinomas are the chromophobe and the collecting duct carcinoma. The chromophobe RCC originates from intercalated cells in the distal convoluted tubule, constitutes about 5–7% of tumors, and it is the subtype with the most favorable outcome (5 years survival rate = 80–100%) [3]. On the contrary, the collecting duct carcinoma, which represents 1–2% of RCC, is very aggressive, and it is associated with the most adverse prognosis [3].

Changes in the molecular mechanisms that regulate cell death are commonly observed in cancer [4,5,6]. Regulated cell death was historically associated with apoptosis [7,8,9] However, increasing evidence indicates that several alternative mechanisms regulate death pathways, such as ferroptosis, pyroptosis, and necroptosis [10,11,12,13,14,15]. Although pharmacological control of cell death pathways was developed in recent years, the identification of these novel death pathways opened additional venues for pharmacological intervention in cancer cells and in the immune system.

A significant percentage of RCC patients (about 30%) undergoing curative surgery for localized RCC develop recurrence or metastases [16]. Therefore, additional novel therapeutic strategies are needed for the treatment of RCC patients, especially for those with advanced and metastatic diseases.

## 2. Programmed Cell Deaths in Renal Cancers

Apoptosis, necroptosis, ferroptosis, and pyroptosis recently emerged as regulated cell death modalities that execute their death program following distinct molecular pathways. Defining how and whether these mechanisms exert a role in pathological conditions, and whether interconnectivity of the signalling and modularity of their execution occurs, is crucial from a therapeutic standpoint.

Greater knowledge of the mechanisms of cell death in kidney tumors is of fundamental importance for identifying new therapeutic targets and/or for prevention strategies, and the identification of new diagnostic and prognostic biomarkers [17,18].

To this end, in this review, we cover progress on major, recently emerged cell death modalities, emphasizing their potential clinical and therapeutic implications. In particular, the most recent scientific evidence concerning the role of apoptosis, ferroptosis, pyroptosis, and necroptosis in renal cancers was selected and discussed, also emphasizing their possible therapeutic potential.

## 3. Apoptosis Regulation in Renal Cancers

Apoptosis is the best-known programmed cell death modality that occurs physiologically during development and aging, and as a homeostatic mechanism to maintain cell populations in tissues. It is generally characterized by distinct morphological characteristics and energy-dependent biochemical mechanisms. Dysregulation of the apoptosis is observed in several human conditions, such as neurodegenerative diseases, ischemic damage, autoimmune disorders, and many types of cancer [13]. The possibility of modulating the death of a cell is recognized for its immense therapeutic potential. In fact, apoptosis is considered an essential defense mechanism against both the occurrence and progression of several neoplasia, including renal carcinomas.

### 3.1. Main Signaling Pathways

The role of the key regulator of apoptosis, the transcription factor [19] p53, was implied in RCC pathogenesis; less evidence is present on the family members [20] p63 and [21,22] p73. Not only are p53 mutations an infrequent event in RCC, but many of the proteins involved in the regulation of apoptosis, such as Puma, Bax, and the Bcl-2 family, are not mutated or dysregulated at an mRNA level in cohorts of patients from cancer databases, such as the Firehose legacy from the TCGA (cbioportal.com accessed on 15 February 2022), Figure 1.

The lack of significant genomic alterations of the genes involved in the apoptosis machinery determines two key features of RCC: (i) apoptosis is impaired mostly due to the alteration of the protein levels of key apoptosis-related proteins by regulatory mechanisms, such as the over-expression of apoptosis inhibitors (an example being the apoptosis inhibitor factor (AIF)), which impacts on the prognosis and reduced postoperative survival of RCC [25], and (ii) the machinery being intact from a genomic point of view; most of the effector proteins of apoptosis are wild-type and pharmacological intervention can be exploited to re-activate this form of cell death. As an example, the modulation of the expression of the protein p21, a known transcriptional target of p53, can be downregulated through the administration of oligonucleotides to renal cancer cell lines, increasing the apoptotic response to classic chemotherapeutic agents, such as doxorubicin and cisplatin [26]. In the context of RCC, the activity of known regulators of apoptosis, such as Bcl-2, whose expression is usually not dysregulated in kidney neoplasms, can be modulated by pharmacological intervention, such as by the inhibition of cyclin-dependent kinase phosphates, such as cdc25. The chemical inhibition of cdc25 in RCC cell lines proved to be effective in restoring apoptosis through the downregulation of Bcl-2 and Bcl-xL, and the upregulation of Bax, Figure 2 [27].

One of the most described mechanisms through which apoptosis is triggered is represented by the tumor necrosis factor-related apoptosis-inducing ligand (TRAIL) [28,29]. TRAIL is a protein showing high homology to CD95, TNF, and Ltα, which are recognized ligands of the tumor necrosis factor receptor (TNFR). Different from the direct activation of the TNFR, which triggers apoptosis in a wide variety of cells and cell lines, TRAIL shows selective activity in transformed cells. According to this, it can be considered a potential selective target to impair cancer cell growth, minimizing side effects on normal cells. Such a mechanism was described in the context of RCC, in a mouse model of metastatic dissemination of Renca cells in BALB/c mice. Using monoclonal TRAIL-inactivating antibodies, a marked increase in liver, but not lung, metastasis in BALB/c mice injected with Renca cells is demonstrated. Moreover, TRAIL^−/−^ mice show increased levels of liver metastasis, if compared to wild-type TRAIL mice. This evidence highlights the role of the apoptosis induced by TRAIL in the regulation of the metastatic potential of renal cancer cells, in a compartment-dependent fashion in vivo [30].

TRAIL involvement in the induction of apoptosis was further studied in RCC, and many compounds seem to be promising agents in the control of renal cancer cell growth, by modulating its activity. IITZ-01, an autophagy inhibitor acting on lysosomes formation, can sensitize RCC cell lines to TRAIL-induced apoptosis, by promoting DR5 upregulation and downregulation of survivin, through a proteasome-dependent degradation. [31] TRAIL seems to be a plausible mechanism through which kidney cancer cells can be induced to die from programmed cell death, with a few molecular mechanisms underpinning its potential. As an example, NF-κB is shown to be over-expressed in many RCC cell lines, and its activation correlates to resistance to TRAIL-induced apoptosis. This is proven by the adenoviral expression of NF-κB inhibitor IκBα, which lowers the shuttling of NF-κΒ to the nucleus of the cell, and restores sensitivity to TRAIL in a wide panel of RCC cell lines, as shown in Figure 2 [32]. Apart from the death receptors, other signaling pathways and molecules are shown to regulate apoptosis in RCC, usually impairing the process [33]. The activation of c-Jun NH2-terminal kinase (JNK) by cordycepin results in increased apoptosis of kidney cancer cells, whereas its inhibition by the selective chemical inhibitor SP600125 increases the nuclear translocation of the β-catenin transcription factor, and the downregulation of Dickkopf-1, a known inhibitor of the Wnt signaling, and a marked decrease in Bax, resulting in diminished apoptotic cell death [34].

The attenuation of apoptosis in RCC may also depend on the transcription of proteins under the direct control of the hypoxia-inducible factor (HIF) [35,36] which is a known mediator of RCC development. Under this light, the apoptosis repressor with a CARD domain (ARC) protein, is actively transcribed after the accumulation of HIF-1 in renal cancer cells. The ARC gene is transcribed after the binding of HIF1 to chromatin in a region upstream of the gene, harboring a hypoxia-responsive element. Therefore, the accumulation of HIF-1, which is not degraded due to the lack of an effective von Hippel–Lindau (VHL) protein (whose gene is one of the most frequently mutated in RCC) increases ARC expression, leading to the inhibition of apoptosis [37]. A controversial role is also described for the activity of HIF-1 towards the activation or inhibition of apoptosis. One of its subunits, HIF-1α is regulated at a protein level by the factor-inhibiting HIF (FIH), which promotes renal cancer cell proliferation by protecting cells from HIF-1α-mediated apoptosis, suggesting that HIF plays contradictive roles in the regulation of cell death (Figure 2) [38].

Most of classical chemotherapeutic agents induce apoptosis in cancer cells, but this process is somehow impaired in RCC, especially in the metastatic setting, describing the disease as resistant to chemotherapy. One of the compounds often used in combination therapy in RCC, with very limited efficacy, is oxaliplatin. This might be due to the anti-apoptotic effect of cFLIP, which counteracts the apoptosis induced by oxaliplatin, which requires caspase activation and reactive oxygen species (ROS) accumulation, prevented by increased XIAP protein accumulation and the constant activation of Akt, mediated by cFILP in RCC [39].

ROS accumulation is a critical regulator of the mitochondrial pathway of apoptosis [40], often prevented in RCC. Therefore, compounds acting on ROS generation show a potential effect on the induction of apoptosis of kidney cancer cells. Compounds such as escin, a natural pentacyclic triterpenoid, induce G2/M phase cell cycle arrest and ROS accumulation, leading to apoptotic cell death in kidney cancer cells, which is abrogated by anti-oxidative treatment with *N*-acetyl cysteine, therefore, confirming the role of ROS in inducing mitochondrial membrane instability, triggering apoptosis [41,42,43,44]. A similar mechanism converging on ROS accumulation is demonstrated in the context of the treatment with a chalcone compound, broussochalcone A (BCA). This compound, when tested on RCC cell lines A498 and ACHN, proved to be effective in upregulating pro-apoptotic proteins, such as p53, Bax, p21, and the active forms of caspase-3 and caspase-7 (diminishing their pro-active forms), and in downregulating the anti-apoptotic Bcl-2 and Bcl-xL. Moreover, BCA treatment led to increased nuclear translocation of the FOXO3 transcription factor, due to impaired DNA reparation and increased ROS generation, thus, suggesting ROS accumulation and FOXO3 transcriptional program activation as possible mediators of BCA-induced apoptosis in RCC cell lines, as shown in Figure 2 [45]. As a possible therapeutical approach to kidney cancer, the sensibilization to TRAIL shows promising results, at least in in vitro experimental settings. A classical chemotherapeutic agent such as paclitaxel, a microtubule formation inhibitor, was tested on RCC cell lines, and low doses show increased production of ceramide, derived from sphingomyelin hydrolysis, which impairs Akt phosphorylation. Decreased levels of Akt correlates with sensitivity to TRAIL, specifically through the suppression of the anti-apoptotic cFLIP [46,47].

Many more compounds show efficacy in RCC cells in vitro, through the modulation of key proteins involved in the regulation of apoptosis, such as Akt, Bax, Bcl-2, and Bcl-xL. An example is represented by norcantharidin, inducing endoplasmic reticulum stress, increasing Bax, and decreasing Bcl-2 and Mcl-1 protein levels [48]; or by the inhibition of 5O-lipoxygenase with a novel 2,5-dihydroxycinnamic acid-based inhibitor, inducing apoptosis, as well as increasing the autophagic flux on RCC4 cells [49]; or, finally, by neferine, an alkaloid extract, which induces apoptosis by the inhibition of the pathway of NF-κB, through the caspase-dependent cleavage of p65 and the consequent downregulation of Bcl-2 [7]. The epigenetic landscape of RCC was also correlated to the attenuation of the apoptotic machinery. RCC usually shows high DNA methylation levels, also impacting on prognosis. Through orthotopic injection of cancer cells in mice, upregulation of DNA methyltransferase 3B (DNMT3B) is detected and confirmed in advanced RCC cases of the clear cell histotype. Through the analysis of the methylated target regions of DNMT3B, the gene encoding a subunit of complex III of the mitochondrial respiratory chain (UQCRH) is highlighted as heavily methylated, and its silencing dramatically increases the formation of primary tumors in vivo, through the decrease in apoptosis mediated by the mitochondrial pathway [50].

### 3.2. Possible Therapeutical Approaches

The role of apoptosis in kidney cancer is still quite controversial, although in vitro and in vivo evidence suggests its potential for a therapeutical approach. At a clinical level, a haplotype-based method for single nucleotide polymorphism analysis shows that regions of the genome associated with RCC, and one of these is represented by the caspase 1-5-4-12 region. Patients harboring this specific haplotype show higher risk of RCC development, also identifying specific CASP1 and CASP5 variants associated with enhanced kidney cancer risk [7,51].

Moreover, pharmacological agents for RCC treatment increase with the introduction of the immune checkpoint inhibitors in the current therapeutic setting. Immune checkpoints are molecules expressed by the immune cells that interact physically with receptors or ligands expressed by tumoral cells, resulting in anergy from the immune system. In the context of RCC, antibodies blocking programmed cell death 1 (PD1), such as nivolumab [52], or blocking its ligand (PD-L1), such as atezolizumab or avelumab, were approved by regulatory agencies (FDA, EMA), together with an antibody interfering with the activity of a different immune checkpoint molecule, CTLA4 (ipilimumab) [53].

The introduction of these antibodies as therapeutic tools, alone or in combination, positively changed the prognosis of RCC patients, especially in the setting of patients expressing high levels of PD-L1. In this scenario, a possible role of immune checkpoints was characterized, and B7-H1, a subfamily of the B7 protein family, is shown to be highly expressed in RCC, unfavorably impacting on its prognosis, and serving as a biomarker of prognosis [54]. This molecule is also shown to induce T-cells apoptosis, in both in vivo and in vitro models of human cancers such as breast cancer, indicating its role as a possible immune escape, promoted by the tumor cell to survive [55].

## 4. Ferroptosis in Renal Cancers

Ferroptosis is a caspase-3-independent cell death, described for the first time by Dixon in 2012 [56]. This cell death phenomenon is characterized by progressive iron-related lipid ROS accumulation, and differs from other forms of cell death, such as necrosis and apoptosis, both from a morphological and a molecular point of view. Specifically, cells undergoing ferroptosis show mitochondrial shrinkage, with increased membrane density, and a reduction in mitochondrial crests, but no swelling of cytoplasm and other organelles, chromatin condensation, genesis of apoptotic bodies, cytoskeleton fragmentation, or the collapse of the cytoplasmic membrane [57,58].

### 4.1. Main Signaling Pathways

Based on the regulatory molecular mechanisms, ferroptosis is commonly divided into three different categories that involve iron homeostasis, lipid metabolism, and the glutathione peroxidase 4 (GPX4) activity (Table 1).

In addition, two independent ferroptosis inhibitory modalities were recently described: the suppressor protein 1 (FSP1)-CoQ10-NAD(P)H pathway and the tetrahydrobiopterin (BH4)-dihydrofolate (DHFR) pathway. Both these pathways inhibit the phospholipid peroxidation that occurs during ferroptosis, by interacting with the activity of glutathione and GPX4.

Concerning the iron and lipid metabolism, it is demonstrated that the bioaccumulation of iron in the cells may induce the peroxidation of proteins, nucleic acids, and lipids, and generate increasing ROS level by a Fenton reaction (Fe^2^ + H_2_O_2_ → Fe^3+^ •OH + OH^−^) [59]. Accordingly, ferroptosis is frequently induced by transcription factors that increase the expression of iron metabolism inhibitors, such as ferritin light chain (FTL), and ferritin heavy chain 1 (FTH1) [60]. Among molecular mediators, RSL3 and RSL5, two small, lethal compounds, play a key role in the iron-dependent oxidative cell death [61]. These molecules negatively modulate the activity of GPX4, thus, increasing the iron accumulation-related oxidative stress.

Several studies propose p53 as a possible mediator of the complex molecular network involved in the ferroptosis process [62,63]. Indeed, it is known that the activation of p53 induces a significant increase in lipid ROS levels [64]. According to this, the increase in lipid ROS is involved in more than 80% of p53-related cell deaths. However, the latest evidence suggests that p53 has a paradoxical effect on ferroptosis. Specifically, p53 may induce ferroptosis through both the inhibition of solute-carrier family 7 member 11 (SLC7A11), and the overexpression of spermidine/spermine *N*1-acetyltransferase 1 (SAT1) or glutaminase 2 (GLS2) [65]. On the other hand, p53 can inhibit ferroptosis, by upregulating cyclin-dependent kinase inhibitor 1A (CDKN1A) [65]. Also, Requena and colleagues found that ferrostatin-1 treatment, an inhibitor of ferroptosis, significantly reduces the rate of p53-related cell deaths [66].

In recent years, the deregulation of ferroptosis was associated with numerous human pathologies, including cancer [67]. Indeed, this form of regulated cell death can play a fundamental role in suppressing tumor growth.

In the landscape of kidney cancers, RCC shows high susceptibility to ferroptosis [68], whereas the clear cells RCCs (ccRCC) undergo ferroptosis only after the silencing of the genes involved in the glutathione peroxidases expression, GPX3 and GPX4 [68].

The high susceptibility of RCCs to ferroptosis can be explained by the unusual dependence of these cancers on the activity of glutathione to maintain cellular homeostasis [69]. Moreover, the high activity of glutathione in RCCs makes these cancers extensively resistant to both chemo- and radiotherapy treatments [70]. Therefore, an innovative therapeutical approach, focused on the modulation of metabolic processes, mainly lipid peroxidation, could represent a good alternative for treating RCCs. In this context, Yang and colleagues [69] perform an extensive in vitro study, in which the sensitivity to erastin-mediated ferroptosis is investigated in 117 cancer cell lines. This study reveals that RCC and the large B cell lymphomas are particularly susceptible to GPX4-associated ferroptosis [69]. In another investigation, the authors demonstrate that TAZ, a Hippo pathway effector, regulates RCC-ferroptosis by inducing the activity of nicotinamide adenine dinucleotide phosphate (NADPH) oxidase 4 (NOX4) [71].

This evidence underlines the fundamental role of ferroptosis in the progression of RCCs. Hence, the identification of possible molecular targets of ferroptosis in RCCs could open new perspectives in the treatment of this cancer, especially in patients who develop resistance to first-line therapies.

**Table 1 ijms-23-06198-t001:** Main molecules involved in ferroptosis.

Molecules	Biological Function	Type of Programmed Cell Death	References
*glutathione peroxidase 4 (GPX4)*	GPX4, an antioxidant defense enzyme, repairs oxidative damage to lipids, and is a leading inhibitor of ferroptosis.	Ferroptosis	[59]
*RSL3*	Transcription factor that increases the expression of iron metabolism inhibitors, such as ferritin light chain (FTL) and ferritin heavy chain 1 (FTH1).	Ferroptosis	[61]
*RSL5*	Transcription factor that increases the expression of iron metabolism inhibitors, such as ferritin light chain (FTL) and ferritin heavy chain 1 (FTH1).	Ferroptosis	[61]
*p53*	p53 has a paradoxical effect on ferroptosis: p53 may induce ferroptosis through both solute-carrier family 7 member 11 (SLC7A11) inhibition, and spermidine/spermine N1-acetyltransferase 1 (SAT1) or glutaminase 2 (GLS2) overexpression; p53 can inhibit ferroptosis by upregulating cyclin-dependent kinase inhibitor 1A (CDKN1A).	Ferroptosis, apoptosis	[62,63]
*glutathione peroxidase 3 (GPX3)*	GPX3, an antioxidant defense enzyme, repairs oxidative damage to lipids.	Ferroptosis	[68]
*nicotinamide adenine dinucleotide phosphate (NADPH) oxidase 4 (NOX4)*	Multi-subunit enzyme complex that utilizes nicotinamide adenine dinucleotide phosphate to produce superoxide anions and other reactive oxygen species. Excess reactive oxygen species generated by NOX promotes ferroptosis.	Ferroptosis, apoptosis	[71]
*SLC7A11*	SLC7A11 overexpression is associated with the inhibition of ferroptosis, and the consequent increase in RCCs proliferation, migration, and invasion.		[72]

In this scenario, several researchers focus their studies on the identification of novel and reliable markers of RCCs-ferroptosis. A multi-omics study by Fangshi Xu et al. [72] assesses the prognostic value of SLC7A11, a ferroptosis regulatory gene, by survival analysis. The investigation of multiple datasets shows that the upregulation of SLC7A11 genes in RCCs acts as a worse, and independent, prognostic factor. Along with other mechanisms, SLC7A11 overexpression is associated with the inhibition of ferroptosis, and the consequent increase in RCCs proliferation, migration, and invasion. The fundamental role of ferroptosis in the development of RCCs is also demonstrated in an in vivo study of Cheng et al. [72], in which resistance to ferroptosis, induced by ferric nitrilotriacetate intake, makes the A/J strain mice highly prone to the development of RCCs. In particular, the ferric nitrilotriacetate, a form of chelated iron, is able to induce RCCs in more than 60% of treated mice, through the alteration of the chromosomes 4, 12, 13, or 14. The main genes involved in these chromosomal alterations are Cdkn2a/2b and p53.

Concerning ccRCCs, the main ferroptosis-related genes (FRGs) were investigated in two independent cohorts of patients, to both identify their possible prognostic value and explore the cellular and molecular mechanisms of ferroptosis [73]. Among the genes investigated, FRGs, HSPB1, FANCD2, TFRC, RPL8, CARS, CDKN1A, and SLC7A11 seem to be associated with the ccRCC occurrence, whereas ccRCCs show a significant downregulation of the MT1G, CISD1, FDFT1, SLC1A5, GLS2, ATP5G3, ACSL4, and HSPA5 genes.

RCC is a tumor with high sensitivity to ferroptosis, and a consensus clustering analysis on data from the TCGA, with regards to ferroptosis regulators, shows two independent clusters with different expressions of PD-L1 and immune infiltrate. The expression of PD-L1 was, therefore, used to understand a possible co-upregulation with a ferroptosis regulator and cysteinyl-tRNA synthetase (CARS), a described inhibitor of ferroptotic cell death shown to be highly expressed by RCC, correlating with the expression of PD-L1 and a worse prognosis and decreased overall survival, showing a possible connection between ferroptosis and the immune network orchestrated by the immune checkpoint molecules in RCC [74].

Combining analysis of these genes could provide the scientific rationale for developing a prognostic score based on the activation/suppression of ferroptosis. Even though data reported here strongly suggest an association between ferroptosis and ccRCCs, it is not yet possible to establish the exact influence of this phenomenon on ccRCC progression.

### 4.2. Possible Therapeutical Approaches

Ferroptosis inducers, such as erastin, can improve the anticancer effects of several anti-RCC drugs. Likewise, it is hypothesized that, sometimes, erastin induces programmed cell death, and exerts therapeutic effects on cancers by activating the ferroptosis phenomenon. As a result, a better understanding of the role of ferroptosis in renal cancers lays the foundation for improving the pharmacological armamentarium available to clinicians for the treatment of highly resistant renal carcinomas.

Key concepts about ferroptosis are listed in Box 1.

Box 1Ferroptosis.
**Key Points**

**References**

Ferroptosis is characterized by progressive iron-related lipid reactive oxygen species (ROS) accumulation;
[57]
During ferroptosis, mitochondrial shrinkage, with increased membrane density and decreased mitochondrial crests, occurs;
[58,59]
RSL3 and RSL5 mediators negatively modulate the GPX4 activity, increasing the iron-dependent oxidative cell death;
[61]
SLC7A11 increases the RCC progression by modulating the ferroptosis phenomenon;
[72]
Ferroptosis mediators are novel reliable targets for RCC therapy.
[73,74]

## 5. Pyroptosis in Kidney Cancer

The phenomenon of pyroptosis is a more recently identified programmed cell death, which is stimulated by both pathogen infection and non-infectious stimuli [75,76]. It plays a crucial role in the clearance of pathogens towards the activation of inflammasomes and the release of pro-inflammatory cytokines [77]. Among the cell death processes, pyroptosis is defined as a singular form of pro-inflammatory programmed cell death that contributes to the innate immune response [78]. This kind of primary response has two key biological outcomes: the inhibition of intracellular pathogen replications, and the activation of pathogen phagocytosis by immune cells. Due to the activation of acute inflammatory responses, moderate levels of pyroptosis show a good control of pathogen infections. Unfortunately, excessive activation of pyroptosis leads to systemic inflammatory reactions, which can result in damage to normal cells, tissue injury, and organ dysfunction and failure [79,80,81,82,83]. Pyroptosis is mediated by inflammatory caspases (caspase-1, -4, and -5), and nuclear condensation, DNA cleavage, cell membrane pore formation, cell swelling, lysis, and the release of cytoplasmic contents are the events that occur to destroy the integrity of cell membranes and lead to inflammatory necrosis [84,85,86,87].

### 5.1. Main Signaling Pathways

Pyroptosis can follow different mechanisms of activation: the inflammasome-dependent or the inflammasome-independent pathways (Table 2). Among the inflammasome-dependent pathways, is it possible distinguish the canonical and non-canonical pathways; the inflammasome-independent pathways, instead, refer to caspase-3 and granzymes proteases mediated-pathways [75]. The canonical inflammasome pathway consists of the recruitment of inflammasome adaptor, apoptosis-associated, speck-like proteins. This first step leads to the binding between inflammasome sensors and the pro-caspase-1; pro-caspase-1, through self-cleavage, is activated in caspase-1 and cleaves gasdermin D (GSDMD), releasing gasdermin-N domain and forming pores on the plasma membrane [88,89]. The generation of pores allows the enrichment of pro-inflammatory cytokines in the extracellular environment, allowing the release of IL-1β and IL-18 in their mature form as an additional action of active caspase-1 cleavage [89,90,91,92]. The activation of all these signals leads to a vicious circle, in which an inflammatory microenvironment, in turn, recruits other immune cells, amplifying the inflammatory response and feeding the inflammasome [93,94,95]. On the contrary, the non-canonical inflammasome pyroptosis shows a different activation modality, in which caspase-4, -5, and -11 are activated by the binding of intracellular lipopolysaccharide (LPS), through the N-terminal of the caspase recruitment domain (CARD) [96]. The activation of caspases leads to the cleavage of GSDMD in N-GSDMD, which moves to the cell membrane to form membrane pores. Moreover, the maturation of IL-1β and IL-18 pro-inflammatory cytokines occurs because of the assembly of NLRP3 triggered by the K^+^ efflux, caused by N-GSDMD. In this pathway, pyroptotic cell death is mediated by the cleavage of another key protein, pannexin-1, through caspase-11 activation [97]. Cleaved pannexin-1 induces the release of cellular ATP, leading to pyroptosis events mediated by the ion channel P2X7 receptor [97]. Regarding the inflammasome-independent pathways, it is worth noting that caspase-3 stimulates gasdermin-inducing pyroptosis. Indeed, it is demonstrated that chemotherapeutic drugs induce caspase-3-mediated gasdermin E (GSDME) cleavage, leading to the formation of N-GSDME and tumor cells pyroptosis [98,99]. Nonetheless, N-GSDME may activate the canonical inflammasome pathway, promoting the release of IL-1β and IL-18 [75]. In addition, gasdermins could be also cleaved by lymphocyte-derived granzyme proteases. In particular, caspase recruitment (CAR) T-cells secrete granzyme-B (GzmB), which is responsible for the activation of caspase-3 and the cleavage of GSDME in target cells, forming membrane pores. Surprisingly, it is also reported that GzmB and granzyme A (GzmA), contained in cytotoxic lymphocytes, induce pores on the plasmatic membrane, through perforin and the hydrolyzation of non-aspartic acid sites [100]. This evidence reveals how pyroptosis cell death can also be triggered by non-caspase-mediated signals [101].

**Table 2 ijms-23-06198-t002:** Main molecules involved in pyroptosis.

Molecules	Biological Function	Type of Programmed Cell Death	Mechanisms	References
Apoptosis-associated, speck-like protein	Involved in the caspase-1-dependent inflammatory pyroptosis, and is the major constituent of the pyroptosome.	Pyroptosis	Canonical inflammasome pathway	[88,89]
Caspase-1	Pyroptosis inducer through cleavage of gasdermin-D (GSDMD) into the active mature peptides.	Pyroptosis	Canonical inflammasome pathway	[84,89]
Gasdermin D	Precursor of the pore-forming protein, allowing the release of mature interleukin-1 (IL1B and IL18), and triggering pyroptosis.	Pyroptosis	Canonical and non-canonical inflammasome pathway	[88,89]
IL-1β	Involved in the transduction of inflammation downstream during pyroptosis processes, and is released through the gasdermin-D pore.	Pyroptosis	Canonical and non-canonical inflammasome pathway	[75,89,90,91,92]
IL-18	Involved in the transduction of inflammation downstream during pyroptosis processes, and is released through the gasdermin-D pore.	Pyroptosis	Canonical and non-canonical inflammasome pathway	[75,89,90,91,92]
Caspase 4	Inflammatory caspase able to promote pyroptosis through NLRP3 and NLRP6 inflammasomes and GSDMD cleavage, in response to non-canonical inflammasome activators.	Pyroptosis	Non-canonical inflammasome pathway	[84,85,86,87,96]
Caspase 5	Responsible for starting pyroptosis through cleavage of GSDMD and the consequent pore formation.	Pyroptosis	Non-canonical inflammasome pathway	[84,85,86,87,96]
Gasdermin E	Pore-forming protein able to both convert non-inflammatory apoptosis to pyroptosis, or promote granzyme-mediated pyroptosis.	Pyroptosis	Non-canonical inflammasome pathway	[98,99]
Caspase recruitment domain (T cells)	It mediates inflammasome activation, and leads to subsequent pyroptosis of CD4+ T-cells and macrophages	Pyroptosis	Non-canonical inflammasome pathway	[96]
NLRP3	It initiates the formation of the inflammasome complex in response to pathogens and damage-associated signals.	Pyroptosis	Non-canonical inflammasome pathway	[97]
Pannexin-1	It leads to channel opening and extracellular ATP release, which, in turn, activates P2X7 receptors and causes cytotoxicity.	Pyroptosis	Non-canonical inflammasome pathway	[97]
LPS	LPS activates phagocytosis-related NADPH oxidase, and leads to the initiation of ROS and NLRP3 inflammasome formation.	Pyroptosis	Non-canonical inflammasome pathway	[96]
P2X7 receptor	Through the formation of membrane pores and K+ efflux through the P2X7-dependent pore, the intracellular Ca2+ concentration increases and ATP-dependent lysis of cells occurs.	Pyroptosis	Non-canonical inflammasome pathway	[97]
Caspase-3	Primary protein responsible for GSDME cleavage and activation, playing an essential role in pyroptosis.	Pyroptosis	Inflammasome non-dependent pathway	[75,98,99]
Gasdermin-E	Precursor of the pore-forming protein.	Pyroptosis	Inflammasome non-dependent pathway	[98,99]
Granzyme-B	Protease delivered into target cells to catalyze cleavage of GSDME and activate caspase-independent pyroptosis.	Pyroptosis	Inflammasome non-dependent pathway	[100,101]
Granzyme A	Protease delivered into target cells to catalyze cleavage of GSDMB and activate caspase-independent pyroptosis.	Pyroptosis	Inflammasome non-dependent pathway	[100,101,102]

Pyroptosis, a lytic cell death process due to its pro-inflammatory nature, causes the activation of a chronic inflammation, which can influence and feed different types of diseases linked to inflammation. An increasing number of studies propose pyroptosis as a crucial event in cancer evolution. Although only specific tumor tissues show pyroptosis pathways involved in angiogenesis, invasion, and metastatic events, some other studies suggest pyroptosis cell mechanisms as a tumorigenesis suppressor, highlighting an alternative approach to kill cancer cells [99,102,103,104,105]. Thus, pyroptosis plays a dual role during tumor progression [106]. Current research reveals that pyroptosis is also involved in the occurrence and development of kidney diseases and kidney cancer, emphasizing their relationship, and questioning the potential therapeutic targets relating to this cell death mechanism [7,107,108,109,110,111]. Kidney cancer is characterized by uncontrolled cell proliferation, the absence of cell death, and different sensitivity to conventional radio- and chemotherapies [112].

In ccRCC, it is still unclear and poorly explored how pyroptosis-related genes interact with each other; however, it is also confirmed that the expression of most pyroptosis regulatory genes is also positively correlated with ccRCC prognosis [111]. Indeed, four pyroptosis regulators were identified in ccRCC tumorigenesis, showing an association between their enhanced expression and poor kidney cancer prognosis [111]. Thus, it is proposed that the activated form of AIM2, an innate immune receptor localized in the cytosol, triggers the caspase-1 pyroptosis pathway, through the recognition of released double-stranded DNA (dsDNA) during pathogenic processes [113]. The involvement of AIM2 in different cancers is already demonstrated; nonetheless, Zhang et al. report an unequivocal cancer-promoting role in ccRCC [111,114,115,116,117]. Also, caspase-5, a non-canonical inflammasome pyroptosis protein member, involves innate immunity in kidney cancer evolution and progression, as well as GZMB, which is used by cytotoxic lymphocytes as protection against malignant cells [118,119]. Another pyroptosis regulator is represented by NOD2; this pattern recognition receptor shows its interaction with extracellular histone H3 induced by LPS, and via the VSIG4/NLRP3 pathway, causing pyroptosis [120,121]. In addition, the study performed by Tang et al. sheds light on FOXD2-AS1 as a new possible therapeutic target, because of its impact on GSDMB and NLRP1, key genes of pyroptosis [109,122]. This investigation also reveals clinicopathological manifestations of pyroptosis-related lncRNAs; specifically, 14 long noncoding RNAs (lncRNAs), AP000553.2, AC022126.1, LINC00941, AL162586.1, SNHG12, AC007743.1, AC099850.3, AL031670.1, FOXD2−AS1, AC015819.2, AC027271.1, MUC12−AS1, LINC02747, and RAP2C−AS1, are shown to be independent prognostic factors for ccRCC [109].

### 5.2. Possible Therapeutical Approaches

From a therapeutic perspective, it is observed that some patients with kidney cancer develop different resistance to the most widely used drugs, such as sunitinib and axitinib, suggesting that different correlations between pyroptosis and tumor-immune microenvironments occur [109,123,124]. Moreover, different tumoral microenvironment-infiltrating cell populations are already considered an unfavorable prognostic factor for ccRCC. However, the prognostic and predictive value of pyroptosis is still not established [125,126]. Starting from the intrinsic heterogeneity of tumors, and their immune microenvironment complexity, the balance between pyroptosis-induced tumor growth and pyroptosis-induced tumor suppression should be further explored, in order to identify new cancer immunotherapies able to prevent and treat kidney cancers [101].

A summary of the key points on pyroptosis are reported in Box 2.

Box 2Pyroptosis.
**Key Points**

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

Enhanced expression of AIM2, caspase-5, NOD2, and FOXD2-AS1 pyroptosis regulators in ccRCC are associated with a poor kidney cancer prognosis.
[109,111,113,114,115,116,117,120,121,122]

## 6. Necroptosis in Kidney Cancer

Necroptosis, a regulated form of necrosis, belongs to processes of programmed cell death, and it is involved in human diseases [12,127]. Similar to other forms of regulated cell death, necroptosis reports mitochondrial membrane permeability transition-dependent necrosis [128,129]. On the contrary, necroptosis also shares some morphological characteristics with accidental cell death [129]; indeed, the swelling of organelles, disruption of plasma membrane, cell lysis, and the release of cellular components which, in turn, activate a pro-inflammatory response, are events that occur during necroptosis [130,131,132,133]. As a secondary effect, the necroptotic cell death also induces an adaptive immune response, triggering the release of pro-inflammatory cytokines, and the subsequent onset of pathologies [128]. However, the scenario is not always so clear; indeed in some diseases, necroptosis demonstrates a protective role [134,135].

### 6.1. Main Signaling Pathways

Different types of stimuli can trigger necroptosis, such as tumor necrosis factor α (TNFα), interferon-γ (IFN-γ), lipopolysaccharide (LPS), and pathogen/damage molecular patterns [136] (Table 3). These stimulating factors are in common with the apoptotic pathway; nonetheless. additional regulators are required to specifically trigger necroptosis [127]. Two main regulators orchestrate the necroptotic process: on one hand there are distinct proteins, such as receptor-interacting protein kinases 1 (RIPK1), receptor-interacting protein kinases 3 (RIPK3), and pseudokinase mixed lineage kinase domain-like proteins (MLKL), which represent the core of necroptosis; on the other hand, post-transcriptional modifications, such as phosphorylation and ubiquitination, necessarily contribute to regulate the necroptotic mechanism [131,137,138,139].

Necroptosis is started by key signalling regulators, such as tumor necrosis factor receptor-1 (TNFR1). Precisely, the binding between TNFR1 and TNF-α enrolls several proteins (RIPK1, TNFR-associated death domain (TRADD), TNFR-associated factor 2 (TRAF2), TNFR-associated factor 5 (TRAF5), linear ubiquitin chain assembly complex (LUBAC), E3 ubiquitin ligases cellular inhibitor of apoptosis 1 (cIAP1), and E3 ubiquitin ligases cellular inhibitor of apoptosis 2 (cIAP2)), to form a high-molecular-weight membrane-associated complex, called *complex I* [140,141,142]. Then, two consequential events occur: LUBAC, together with cIAP1 and cIAP2 in *complex I*, causes the Lys63 residue polyubiquitination of RIPK1 which, in turn, induces the recruitment of the IkB kinase complex (composed of IKKα, IKKβ, and NEMO) and TAK1 complex (consisting of TAK1, TAB1, and TAB2) [143]. TAK and IKK complexes lead to the activation of the nuclear transcription factor-kappa beta (NF-kB), allowing the cell survival through the expression of pro-inflammatory and pro-survival genes [144]. Briefly, *complex I* activates the NF-kB pathway to promote cell survival.

Moreover, the formation of new complexes, *complex IIa* or *complex Iib*, are induced when RIPK1 is deubiquitinated by the cylindromatosis enzymes (CYLD), while A20, RIPK1, and TRADD are separated from TNFR1 [145,146,147]. *Complex IIa* consists of TRADD, FADD, and RIPK1, which recruit and activate pro-caspase-8 in caspase-8 through its cleavage. The activation of caspase-8 causes apoptosis without RIPK1 kinase activity [145,147]. *Complex IIb*, instead, takes place in the absence of cIAP1/2, with the inhibition of TAK1 or IKK complexes, and acts only through FADD, RIPK1, and pro-caspase-8 [148]. In this case, the cleavage of pro-caspase-8 in caspase-8 occurs due to RIPK1 kinase activity, leading to RIPK1-dependent apoptosis [127]. Therefore, the necroptosis pathway is activated only under these conditions: high levels of both RIPK3 and MLKL, and absent or inactivated caspase-8. Thus, the involvement of RIPK3, together with RIPK1 and MLKL, induces the crucial event of necrosome formation, causing necroptosis [146,147].

Necroptosis is mediated by RIPK1, RIPK3, and MLKL. Through *complex IIa or IIb*, RIPK1 leads to cell death-inducing apoptosis; however, if RIPK1 is hyperphosphorylated by TAK1, the necroptotic pathway is triggered [127]. In particular, E3-ligase PELI1 causes the polyubiquitination of RIPK1, allowing its interaction with RIPK3 [149]. It is noteworthy that RIPK1, as a kinase, is capable of auto-phosphorylation, inducing necroptosis differently from the classical way [150]. Although RIPK1 shows a crucial implication in the induction of necroptosis, some studies highlight how necroptosis cannot be fully dependent on RIPK1. Specifically, it is reported that RIPK1 knockdown inhibits TNF-α-induced necroptosis, but it is not able to block necroptosis induced by LPS [151]; in addition, necroptosis can be suppressed by RIPK1, acting on RIPK3, causing its inhibition [152]. The association between RIPK1 and RIPK3 causes homodimerization, and consequent auto-phosphorylation and activation [127]. Phosphorylated RIPK3 leads to the recruitment and phosphorylation of MLKL, inducing necrosome formation [127]. Inhibitors of RIPK3, such as GSK’840, GSK’843, HS-1371, and the metalloprotease 17 (ADAM17), initiate the apoptotic pathway, as well as the de-ubiquitination of RIPK3 by A20 blocks necroptosis, through the activation of lysosomal degradation [153,154]. Lastly, RIPK1 and RIPK3, together with MLKL, form the necrosome, which, under RIPK-3 phosphorylation events, causes the oligomerization of MLKL and the formation of a MLKL octamer [155]. Only at this point can the octamer MLKL be released by the necrosome, and reach the plasma membrane to disrupt its integrity and induce necroptosis [155]. However, also in this case, MLKL inhibitors, such as necrosulfonamide (NSA), thioredoxin-1 (Trx-1), and GW806742X, may act, blocking MLKL phosphorylation, oligomerization, and plasma membrane translocation [156], confirming a crucial redox regulation [157,158].

**Table 3 ijms-23-06198-t003:** Main molecules involved in necroptosis.

Molecules	Biological Function	Type of Programmed Cell Death	References
IFN-γ	Responsible for triggering necroptosis.	Necroptosis	[136,140,141,142]
LPS	Responsible for triggering necroptosis.	Necroptosis	[136,140,141,142,151]
RIPK1	A key regulator of the assembly of complex IIb (RIPK1-RIPK3-MLKL) during necroptosis.	Necroptosis	[127,131,137,138,139,140,141,142,143,145,146,147,148,149,150,151,152,155]
RIPK3	Serine/threonine-protein kinase that activates necroptosis.	Necroptosis	[127,131,137,138,139,140,141,142,146,147,149,152,155]
MLKL	Pseudokinase that plays a key role in TNF-induced necroptosis.	Necroptosis	[127,131,137,138,139,140,141,142,146,147,155]
TNFR1	Its activation allows the recruitment of TRADD, RIP1, and TRAF2.	Necroptosis	[140,141,142,145,146,147]
TRADD	It is identified as a target protein for TNF-induced necroptosis in the absence of RIPK1.	Necroptosis	[140,141,142,145,146,147]
FADD	It recruits the initiator caspase-8, forming the death-inducing signaling complex (DISC).	Necroptosis	[148]
TRAF2	Through TNF-induced NF-κB activation, it is able to protect cells, inhibiting necroptotic cell death.	Necroptosis	[140,141,142]
TRAF5	Through TNF-induced NF-κB activation, it is able to protect cells, inhibiting necroptotic cell death.	Necroptosis	[140,141,142]
LUBAC	It regulates necrosome-associated RIPK1 ubiquitination.	Necroptosis	[140,141,142,143]
cIAP1	It ubiquitinates NF-kB, inducing kinase (NIK) to suppress non-canonical NF-kB signaling and RIPK1 to promote cell survival.	Necroptosis	[140,141,142,143,148]
cIAP2	It ubiquitinates NF-kB, inducing kinase (NIK) to suppress non-canonical NF-kB signaling and RIPK1 to promote cell survival.	Necroptosis	[140,141,142,143,148]
IKKα	Together with IKKβ, it constitutes IkB kinase complex.	Necroptosis	[143,144,148]
IKKβ	Together with IKKα, it constitutes IkB kinase complex.	Necroptosis	[143,144,148]
NEMO	Together with IKKα and IKKβ, it constitutes IkB kinase complex.	Necroptosis	[143,144]
TAK1	Serine/threonine kinase, which phosphorylates RIPK1, regulating its interaction with RIPK3 and promoting necroptosis. It constitutes TAK1 complex.	Necroptosis	[127,143,144,148]
TAB1	Together with TAK1 and TAB2, it constitutes TAK1 complex.	Necroptosis	[143,144]
TAB2	Together with TAK1 and TAB1, it constitutes TAK1 complex.	Necroptosis	[143,144]
NF-kB	Its activation, through TAK1 and IKK complexes, allows the cell survival.	Necroptosis	[144]
CYLD	Deubiquitinase, which induces TNF-alpha-induced necroptosis.	Necroptosis	[145,146,147]
E3-ligase PELI1	Negatively regulates necroptosis by reducing RIPK3 expression.	Necroptosis	[149]

As reported above, the activation of necroptosis is a result of tightly regulated processes, without which the uncontrolled inflammatory responses and the consequent onset of human diseases occur [128]. Necroptosis is reported in numerous pathological conditions, including cancers [159]. Among the investigated pathologies, necroptosis also demonstrates its implication in kidney diseases; indeed, both acute kidney injury (AKI) and chronic kidney disease (CKD) report increased levels of RIPK1, RIPK3, and MLKL, suggesting necroptosis as a crucial event in kidney injuries [160,161]. On the contrary, treatments with inhibitors of RIPK1 and/or RIPK3 kinase activities, such as Nec-1 (inhibitor of RIPK1), show a reduced MLKL membrane translocation, and a significantly improved cell viability and renal function [159,160,162,163]. Another study reports an increased expression of and interaction between RIPK3 and MLKL, associated with necroptosis occurrence in renal tubular cells, promoting the activation of the inflammasome during ischemia-reperfusion injury (IRI) condition [164]. The same study also highlights the protective role of RIPK3 and MLKL knockout genes on renal tubular cells from inflammasome activation, fibrosis, and necroptosis [164]. Occasionally, kidney injuries may be accompanied by crystal formations which, in turn, seems to activate the necrosome: decreasing the expression of RIPK3 and MLKL, cytotoxicity induced by crystals is inhibited [165]. Sureshbabu et al. also demonstrate that RIPK3-deficient, but not MLKL-deficient, mice are resistant to kidney tubular injury, and the protection induced by RIPK3 deletion is evaluated, considering reduced indicators of oxidative stress, such as nitrotyrosine, and expression levels of mitochondrial NOX4 [166]. Thus, the authors conclude that, independently from MLKL, RIPK3 may promote kidney tubular injury through mitochondrial dysfunction, identifying a promising therapeutic target [166]. Nonetheless, the study of Jing et al. sheds light on the controversial role of RIPK3 in different malignant lesions: in colon cancer, the lack of RIPK3 causes the activation of the NF-kB pathway, inducing increased levels of IL-6 and tumor progression [167]. Even if this scenario suggests a protective role of RIPK3 against tumorigenesis, in cervical cancer, elevated expression of RIPK3 is detected [168]. In the context of RCC, necroptosis is less characterized, as compared to the other forms of cell deaths. A study conducted by Lamki et al. identifies a subpopulation of high-grade RCC that show increased expression of both RIPK1 and RIPK3. This population undergoes cell death after the treatment with TNF mutein, a TNFR agonist that activates the isoform 1 rather than 2. After the treatment, very few cells activate apoptosis with caspase-3 cleavage, while the majority die by necroptosis. This is explained by the induction of the expression of RIPK1 and RIPK3, and the consequent phosphorylation of MLKL and Drp1, leading to the MLKL octamer formation and migration to the plasmatic membrane [169]. Moreover, necroptosis can be re-activated in RCC by the inhibition of the NF-κB signaling, which is partially obtained by the use of the proteasome inhibitor PS-341; this inhibition leads to IFN-γ triggered necroptosis through JAK1/2 and STAT1 signaling [170]. Necroptosis is also shown to be potentially reactivated in RCC by treatment with emodin, a Chinese medicinal herb, which increases phosphorylation levels of RIP1 and MLKL while increasing the accumulation of reactive oxygen species, responsible for the JNK activation of necroptosis itself [171]. Necroptosis is also impaired in the context of RCC by the expression of specific miRNAs. As examples, miR-124, often over-expressed in RCC cells compared to the normal adjacent tissue, reduces sensitivity to cisplatin by lowering the levels of CAPN4 [172], while miR-381-3p impairs both apoptosis and necroptosis by suppressing TNF activation of both modalities of cell death [173].

### 6.2. Possible Therapeutical Approaches

Starting from the above consideration, many kinase inhibitors were investigated as new possible drugs, and some RIPK1 and RIPK3 inhibitors were already tested and implicated in clinical practice [174]. GNF-7, previously identified as a kinase inhibitor for Bcl–Abl, demonstrates its efficacy as a dual kinase inhibitor for RIPK1 and RIPK3 and necrosome formation. Qin et al. also highlight the more potent capacity of GNF-7 to inhibit necroptosis than the widely used RIPK1 inhibitor Nec-1 [175]. Other inhibitors include RIPK1-specific necrostatins, RIPK3-targeting molecules (GSK’872, GSK’843, dabrafenib), and the necrosulfonamide, which targets MLKL [175,176,177,178].

Since heterogeneous subclones in RCC often cause multidrug resistance to targeted therapies, new personalized drugs based on cell death-related inhibitors, specifically designed for the expression profile of molecular markers, may be fundamental to increase the survival rate of patients with kidney cancer [7].

The principal concepts of necroptosis are highlighted in Box 3.

Box 3Necroptosis.
**Key Points**

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

## 7. Conclusions

Renal cell cancer was considered an orphan disease from a therapeutic point of view, up to the moment at which the increased knowledge on its pathogenesis, and the discovery of the role played by the mutation of VHL on the stabilization of HIF and the subsequent activation of the neoangiogenetic program, led to the development of small molecule target agents directed against the vascular endothelial growth factor (VEGF) or its receptor (VEGFR). The introduction of these therapies in the current treatment of the metastatic disease improves the survival of patients. After the development of the anti-VEGF therapeutic agents, the role of mTOR was also evaluated and used as a target for advanced RCC, with limited benefit. RCC patients could access different lines of therapy which could guarantee, in selected cases, prolonged stabilization of the disease. Those treatments, independently from the sequence chosen, inevitably fail in most cases. Therefore, understanding on the molecular basis of the disease [179,180,181,182], and the way in which cell death could be triggered [183,184,185] in RCCs, is still a valuable field of research. Programmed cell death modalities, which form apoptosis to necroptosis, are normally kept inactive in the development of RCCs, to favor cancer cell growth. Mutations in genes involved in these pathways are rarely seen, suggesting that their inhibition is achieved mostly by the regulation of the expression of the involved proteins, or by the regulation of their activity. Therefore, the molecular machinery responsible for programmed cell death can be reactivated by pharmacological intervention, and this assumption increases their potential as possible therapeutic targets. Among the described modalities of programmed cell death, apoptosis and ferroptosis show promising results. Regarding apoptosis, many compounds were developed for its regulation, and were tested in a wide variety of cancers. The exploration of their effect on RCC, also in combinatory therapy with other effective molecules or with immunotherapy, might be potentially beneficial. Ferroptosis itself seems to be a prominent cell death modality in both normal kidney and RCCs, and, therefore, its pharmacological regulation might be of extreme importance in this context. Numerous studies suggest that the resistance to programmed cell death is one of the main factors related to both RCCs occurrence and progression. Therefore, targeting programmed cell death pathways for the treatment of RCCs could be therapeutically important. Both the targeting of intrinsic and extrinsic signalling involved in RCC programmed cell death, as well as interfering with other associated cell death mechanisms, currently represent innovative anticancer strategies. As reported in this review, there are several drugs tested in clinical trials, both as a monotherapy and in combination with chemotherapy, radiation therapy, or other inhibitors. Nevertheless, it is very challenging to modulate programmed cell death pathways in RCCs by targeted molecules, thus, constraining the formation of cancer resistance.

## Figures and Tables

**Figure 1 ijms-23-06198-f001:**
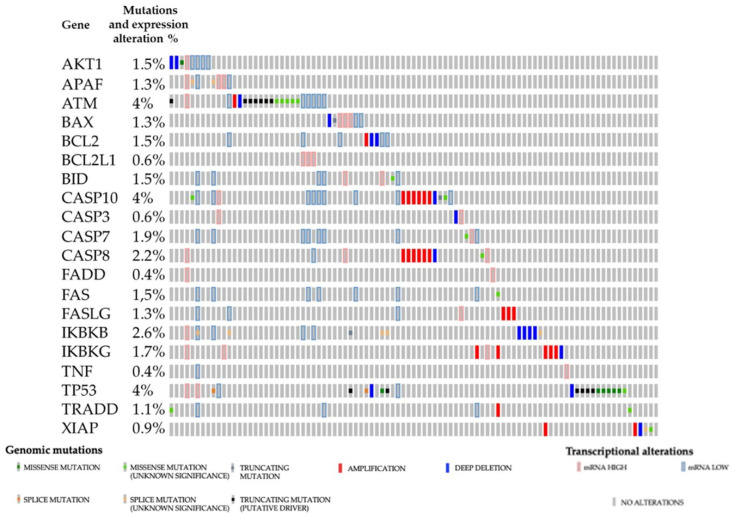
Analysis of mutations and gene expression alterations of apoptotic genes from the Firehose legacy database, accessed through cBioportal [23,24]. Apoptosis-related genes are rarely mutated in renal cell carcinomas.

**Figure 2 ijms-23-06198-f002:**
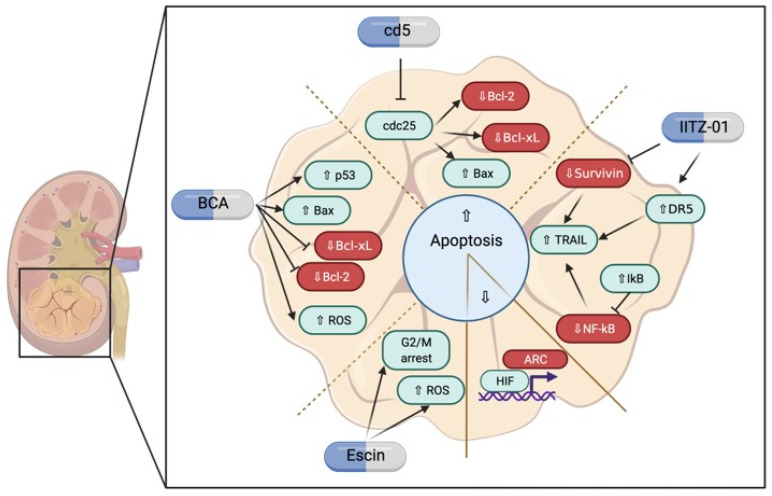
Apoptosis regulation in kidney cancer. The scheme highlights the main regulators of the apoptosis in renal cell carcinoma, as well as showing possible pharmacological interventions for its modulation.

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
