# Peer review of "No Time to Die: How Kidney Cancer Evades Cell Death"

_ijms, 2022, doi:10.3390/ijms23116198_

Round 1

Reviewer 1 Report

In this review article, Ganini and colleagues are summarizing the molecular features implicated in the different mechanisms of cell death in renal carcinoma as well as the pharmaceutical intervention tested so far in vitro and in vivo. The review article is nicely written, there is a good flow and reads well. Here are some minor comments/suggestions:

1) While reading, I had the sense that more illustration is needed. Could you please provide some figures that may help the reader to better understand and follow what is written in the text? For instance, could you present the molecules described in the text within the pathways and indicate the compound used for intervention? Furthermore, this would also allow for checking the interactions among the different signaling pathways. Alternatively, a table summarizing the compounds tested so far, the targeted protein/feature, the targeted pathway along with the function of the compound and the molecules could be also helpful.

2) What is the progress so far with immune checkpoint inhibitors on cell death in renal carcinoma? Could you please describe in the appropriate section?

3) With regards to apoptosis regulation in renal cancers (section 2) you mention that many proteins involved in the regulation of apoptosis are not mutated or dysregulated at the mRNA level. Then how is apoptosis dysregulated in renal carcinoma and thus pharmacologically targeted? Is this a matter of their expression at the protein level or their activity? Please describe it a little better to avoid any confusion of the readers.

4) Also, in page 5, the text described in lines 154-159 is not illustrated in Figure 2 (as indicated). Please revise this.

5) Please provide a better version of Figure 1.

6) At the end, could you please provide your view on how close we are to treat renal carcinoma by targeting the mechanisms of cell death? How close are we to clinical practice and what are biggest challenges to face?

Author Response

Ref: ijms-1690369

" No Time to Die: how Kidney Cancer evades cell death "

 Submitted to: International Journal of Molecular Science

Before we begin the point-by-point review of the list of concerns, we would like to thank the Reviewer for their comments on how to improve the manuscript, which has been revised accordingly, as well as the Editors for calling for a new submission of an improved version of our manuscript.

Reviewer#1

In this review article, Ganini and colleagues are summarizing the molecular features implicated in the different mechanisms of cell death in renal carcinoma as well as the pharmaceutical intervention tested so far in vitro and in vivo. The review article is nicely written, there is a good flow and reads well. Here are some minor comments/suggestions:

Reply: we would like to thank the Reviewer for expressing interest in our work, and for their availability to review our manuscript.

While reading, I had the sense that more illustration is needed. Could you please provide some figures that may help the reader to better understand and follow what is written in the text? For instance, could you present the molecules described in the text within the pathways and indicate the compound used for intervention? Furthermore, this would also allow for checking the interactions among the different signaling pathways. Alternatively, a table summarizing the compounds tested so far, the targeted protein/feature, the targeted pathway along with the function of the compound and the molecules could be also helpful.

Reply: Thanks for this point out. In the new version of our manuscript, we modified both figure 1 and 2. Also we added 3 tables that summarized the main molecules involved in Ferroptosis, Pyroptosis and Necroptosis.

What is the progress so far with immune checkpoint inhibitors on cell death in renal carcinoma? Could you please describe in the appropriate section?

Reply: Effectively, the therapeutical approach to RCC has been revolutionized in the last 10 years by the introduction of immune checkpoint inhibitors. We thank the Reviewer for the suggestion and introduced text regarding some highlights on the link between immune checkpoints and apoptosis and ferroptosis in the pertinent paragraphs. The text now reads:

Moreover, pharmacological agents for RCC treatment have been increased by the introduction of the immune checkpoint inhibitor in the current therapeutic setting. Immune checkpoints are molecules expressed by the immune cells which interact physically with receptors or ligands expressed by tumoral cells, resulting in anergy from the immune system. In the context of RCC, antibodies blocking Programmed Cell Death 1 (PD1) such as Nivolumab or blocking its ligand (PD-L1) such as Atezolizumab or Avelumab have been approved by regulatory agencies (FDA, EMA) , together with an antibody interfering with the activity of a different immune checkpoint molecule CTLA4 (Ipilumumab).

The introduction of these antibodies as therapeutic tools, alone or in combination, has changed positively RCC patient’s prognosis, especially in the setting of patients expressing high levels of PD-L1. In this scenario, a possible role of immune checkpoints has been characterized, and B7-H1, a subfamily of the B7 protein family, has been shown to be highly expressed in RCC, impacting unfavourably on its prognosis and serving as a biomarker of prognosis. This molecule has also been shown to induce T-cells apoptosis both in vivo end in vitro models of human cancers such breast cancer, indicating its role as a possible immune escape promoted by the tumor cell to survive.

3) With regards to apoptosis regulation in renal cancers (section 2) you mention that many proteins involved in the regulation of apoptosis are not mutated or dysregulated at the mRNA level. Then how is apoptosis dysregulated in renal carcinoma and thus pharmacologically targeted? Is this a matter of their expression at the protein level or their activity? Please describe it a little better to avoid any confusion of the readers.

Reply: We thank the Reviewer for this thorough suggestion. The sentence has been reshaped to explain its original meaning: the apoptosis machinery is intact from a mutational point of view and can be exploited pharmacologically. It is impaired in RCC due to alteration of the regulation of the protein expression or activity of some of the key effectors. The text now reads:

Lack of significant genomic alterations of the genes involved in the apoptosis machinery determines two key features of RCC: i) apoptosis is impaired mostly due to alteration of the protein levels of key apoptosis-related proteins by regulatory mechanisms such as the over-expression of apoptosis inhibitors (an example being the Apoptosis Inhibitor Factor -AIF-), which is shown to be impacting on the prognosis and reduced postoperative survival of RCC. Therefore, regulation of the process can be achieved through alterations of its regulation rather than on the downregulation or over-expression of one of its components; ii) being the machinery intact from a genomic point of view, most of the effector proteins of apoptosis are wild-type and pharmacological intervention can be exploited to re-activate this form of cell death.

4) Also, in page 5, the text described in lines 154-159 is not illustrated in Figure 2 (as indicated). Please revise this.

Reply: Extreme apologies for the mistake. The figure has been adjusted with the introduction of Escin and its effect on ROS production and cell cycle arrest. The new figure is the one reported here.

5) Please provide a better version of Figure 1.

Reply: Unfortunately, part of the figure is obtained through the analysis on cBioportal. Anyhow, we reshaped its content, ameliorating the figure legend as well as introducing titles to explain the significance of the percentages reported.

6) At the end, could you please provide your view on how close we are to treat renal carcinoma by targeting the mechanisms of cell death? How close are we to clinical practice and what are biggest challenges to face?

Reply: we would like to thank the Reviewer for this pleasing suggestion. According to this, we have deeply revised the text of our manuscript.

Specifically, the following changes have been made

Paragraph “3.2 Possible therapeutical approaches”

Moreover, pharmacological agents for RCC treatment have been increased by the introduction of the immune checkpoint inhibitor in the current therapeutic setting. Immune checkpoints are molecules expressed by the immune cells which interact physically with receptors or ligands expressed by tumoral cells, resulting in anergy from the immune system. In the context of RCC, antibodies blocking Programmed Cell Death 1 (PD1) such as Nivolumab (Motzer RJ 2015) or blocking its ligand (PD-L1) such as Atezolizumab or Avelumab have been approved by regulatory agencies (FDA, EMA) , together with an antibody interfering with the activity of a different immune checkpoint molecule CTLA4 (Ipilumumab) (Atkins MB 2018).

The introduction of these antibodies as therapeutic tools, alone or in combination, has changed positively RCC patient’s prognosis, especially in the setting of patients expressing high levels of PD-L1. In this scenario, a possible role of immune checkpoints has been characterized, and B7-H1, a subfamily of the B7 protein family, has been shown to be highly expressed in RCC, impacting unfavourably on its prognosis and serving as a biomarker of prognosis. (Thompson RH) This molecule has also been shown to induce T-cells apoptosis both in vivo end in vitro models of human cancers such breast cancer, indicating its role as a possible immune escape promoted by the tumor cell to survive.

Paragraph “Ferroptosis in renal cancers”

RCC is a tumor with high sensitivity to ferroptosis and a consensus clustering analysis on data from the TCGA, with regards to ferroptosis regulators, has shown two independent clusters with different expression of PD-L1 and immune infiltrate. The expression of PD-L1 has therefore been used to understand a possible co-upregulation with a ferroptosis regulator and cysteinyl-tRNA synthetase (CARS), a described inhibitor of ferroptotic cell death, has been shown to be highly expressed by RCC, correlating with the expression of PD-L1 and with a worse prognosis as decreased overall survival, showing a possible connection between ferroptosis and the immune network orchestrated by the immune checkpoint molecules in RCC. (Wang, S)

“3.1 Main signaling pathways” paragraph

Lack of significant genomic alterations of the genes involved in the apoptosis machinery determines two key features of RCC: i) apoptosis is impaired mostly due to alteration of the protein levels of key apoptosis-related proteins by regulatory mechanisms such as the over-expression of apoptosis inhibitors (an example being the Apoptosis Inhibitor Factor -AIF-), which is shown to be impacting on the prognosis and reduced postoperative survival of RCC (Wang Z). Therefore, regulation of the process can be achieved through alterations of its regulation rather than on the downregulation or over-expression of one of its components; ii) being the machinery intact from a genomic point of view, most of the effector proteins of apoptosis are wild-type and pharmacological intervention can be exploited to re-activate this form of cell death.

Reviewer 2 Report

The review of Ganini et al. is timely overview of the involvement of  programmed cell death in the pathogenesis of renal cell carcinoma. The Authors comment on the impairment of apoptosis, ferroptosis, pyroptosis and necroptosis in kidney cancer as a factor that facilitates its development and as the therapeutic target.  The paper is interesting and could be recommended for publication in IJMS after a thorough revision. In particular:

  • The layout of the manuscript is chaotic. First, I would strongly advise to provide a separate subheading on the general significance and the types of PCD that would follow the Introduction;
  • Then, the Chapters 2-5 should be revised to clearly distinguish between the signaling pathways involved in the induction of the specific PCD type and the therapeutic ways to interfere with their activity. Perhaps, they could be divided into conclusive sub-headings, each dealing with the specific signaling system or pathway. In the present form Chapters 2-5 are (over)loaded with data but they lack the structure and the “take-home message”;
  • There are numerous sections that should be re-localized. For instance: lines: 39-40 (page 1), lines 207-231 (page 6) and lines 377-384 (page 9) to point out a few.
  • The language should be thoroughly edited. At the moment, the message of the considerable parts of the manuscript is very difficult to follow;

Author Response

Ref: ijms-1690369

" No Time to Die: how Kidney Cancer evades cell death "

 Submitted to: International Journal of Molecular Science

Before we begin the point-by-point review of the list of concerns, we would like to thank the Reviewer for their comments on how to improve the manuscript, which has been revised accordingly, as well as the Editors for calling for a new submission of an improved version of our manuscript.

Reviewer#2

The review of Ganini et al. is timely overview of the involvement of  programmed cell death in the pathogenesis of renal cell carcinoma. The Authors comment on the impairment of apoptosis, ferroptosis, pyroptosis and necroptosis in kidney cancer as a factor that facilitates its development and as the therapeutic target.  The paper is interesting and could be recommended for publication in IJMS after a thorough revision.

Reply: we would like to thank the Reviewer for expressing interest in our work, and for their availability to review our manuscript.

The layout of the manuscript is chaotic. First, I would strongly advise to provide a separate subheading on the general significance and the types of PCD that would follow the Introduction;

Reply: thanks for this point out. According to your suggestion, we provided a separate subheading on the general significance and the types of PCD.

Specifically, the following paragraph was added:

  1. Programmed Cell Deaths in Renal Cancers

Apoptosis, necroptosis, ferroptosis, and pyroptosis have recently emerged as regulated cell death modalities that execute their death program following distinct molecular pathways. Defining how and whether these mechanisms exert a role in pathological conditions and whether interconnectivity of these signalling and modularity of their execution occur is crucial from a therapeutic standpoint.

Greater knowledge of the mechanisms of cell death in kidney tumours appears to be of fundamental importance for identifying new therapeutic targets and/or for prevention strategies and the identification of new diagnostic and prognostic biomarkers [17,18].

To this end, in this review, we covered some recent progress on emerged cell death modalities. In particular, the most recent scientific evidences concerning the role of apoptosis, ferroptosis, pyroptosis and necroptosis in renal cancers were selected and discussed also emphasizing the possible therapeutic aspects associated with them. 

Then, the Chapters 2-5 should be revised to clearly distinguish between the signaling pathways involved in the induction of the specific PCD type and the therapeutic ways to interfere with their activity. Perhaps, they could be divided into conclusive sub-headings, each dealing with the specific signaling system or pathway. In the present form Chapters 2-5 are (over)loaded with data but they lack the structure and the “take-home message”;

Reply: thanks for this point out. According to your suggestion, we divided each programmed cell deaths into three parts: title of the chapter, “Main signaling pathways” and “Possible therapeutical approaches”.

There are numerous sections that should be re-localized. For instance: lines: 39-40 (page 1), lines 207-231 (page 6) and lines 377-384 (page 9) to point out a few.

Reply: thanks for your suggestion. In the new version of our paper, we deeply re-organized the structure of the manuscript by adding several sub-headings.

The language should be thoroughly edited. At the moment, the message of the considerable parts of the manuscript is very difficult to follow;

Reply: we performed a deeply revision of the language.

Round 2

Reviewer 2 Report

I have no more comments